# Irregular sleep habits, regional grey matter volumes, and psychological functioning in adolescents

Winok Lapidaire[1], Anna S. Urrila[1,2,3], Eric Artiges[1,4], Ruben Miranda[1],
Hélène Vulser[1], Pauline Bézivin-Frere[1], Hervé Lemaître[1], Jani Penttilä[1,5],
Tobias Banaschewski[6], Arun L. W. Bokde[7], Uli Bromberg[8], Christian Büchel[8], Patricia
J. Conrod[9,10], Sylvane Desrivières[9,11], Vincent Frouin[12], Jürgen Gallinat[13],
Hugh Garavan[7,14], Penny Gowland[15], Andreas Heinz[13], Bernd Ittermann[13],
Dimitri Papadopoulos-Orfanos[12], Tomáš Paus[16], Michael N. Smolka[17],
Gunter Schumann[7,11], Marie-Laure Paillère Martinot[1,18], Jean-Luc Martinot[1,19]*, for the
IMAGEN consortium[¶]

1 National Institute of Health and Medical Research, INSERM U A10 "Trajectoires développementales & psychiatrie", University Paris-Saclay, Ecole Normale Supérieure Paris-Saclay, CNRS, Centre Borelli, Gif-sur-Yvette, France, 2 Department of Health, Unit of Mental Health, National Institute for Health and Welfare, Helsinki, Finland, 3 Department of Psychiatry / Adolescent Psychiatry, University of Helsinki and Helsinki University Central Hospital, Helsinki, Finland, 4 Psychiatry Department, EPS Barthelemy Durand, Etampes, France, 5 Adolescent Psychiatry Department, Medical School, Tampere University, Tampere, Finland, 6 Central Institute of Mental Health, Medical Faculty Mannheim, Heidelberg University, Mannheim, Germany, 7 Institute of Neuroscience, Trinity College Dublin, Dublin, Ireland, 8 University Hospital Hamburg Eppendorf, Hamburg, Germany, 9 Institute of Psychiatry, Psychology and Neuroscience, King's College London, London, United Kingdom, 10 Department of Psychiatry, University of Montreal, CHU Ste Justine Hospital, Montréal, Canada, 11 MRC Social, Genetic and Developmental Psychiatry (SGDP) Centre, London, United Kingdom, 12 Neurospin, Commission for Atomic and Alternative Energy, Saclay, France, 13 Department of Psychiatry and Psychotherapy, Campus Charité Mitte, Charité – University Medical Centre Berlin, Berlin, Germany, 14 Departments of Psychiatry and Psychology, 6436 UHC, University of Vermont, Burlington, Vermont, United States of America, 15 Sir Peter Mansfield Magnetic Resonance Centre, University of Nottingham, Nottingham, United Kingdom, 16 Holland Bloorview Kids Rehabilitation Hospital and Departments of Psychology and Psychiatry, Bloorview Research Institute, University of Toronto, Toronto, Canada, 17 Department of Psychiatry and Psychotherapy, Technical University Dresden, Dresden, Germany, 18 Department of Child and Adolescent Psychiatry, AP-HP. Sorbonne Université, Pitié-Salpêtrière Hospital, Paris, France, 19 Centre of Neuroimaging Research, CENIR at ICM Institute, Paris Cedex, France

☯ These authors contributed equally to this work.
¶ A list of the IMAGEN consortium collaborators is provided in the Acknowledgments. The lead IMAGEN consortium author for this publication is Prof. Jean-Luc Martinot.
* jean-luc.martinot@inserm.fr

**Data Availability Statement:** Raw data cannot be shared publicly because they comply to the EU General Data Protection Regulation (GDPR), and some participants did not consent to give public

## Abstract

Changing sleep rhythms in adolescents often lead to sleep deficits and a delay in sleep timing between weekdays and weekends. The adolescent brain, and in particular the rapidly developing structures involved in emotional control, are vulnerable to external and internal factors. In our previous study in adolescents at age 14, we observed a strong relationship between weekend sleep schedules and regional medial prefrontal cortex grey matter volumes. Here, we aimed to assess whether this relationship remained in this group of adolescents of the general population at the age of 16 (n = 101; mean age 16.8 years; 55% girls). We further examined grey matter volumes in the hippocampi and the amygdalae, calculated with voxel-based morphometry. In addition, we investigated the relationships between sleep

access to their individual data. Individual data access requests can be made by sending a completed proposal to simon.roux@kcl.ac.uk for circulation to the IMAGEN Executive Committee. For documentation and data access request forms, please see: https://imagen-europe.com/resources/imagen-project-proposal/.

**Funding:** This study was funded by the European Union-funded FP6 Integrated Project IMAGEN (LSHM-CT-2007-037286), agence nationale de la recherche (ANR-12-SAMA-0004 and ANR-19-CE37-0017-03), Eranet (ANR-18-NEUR00002-01–ADORe and AF12-NEUR0008-01 - WM2NA), Fondation pour la Recherche Médicale (DPA20140629802), Mission Interministérielle de Lutte Contre les Drogues et les Conduites Addictives, and Fondation de France (00081242) in the form of grants awarded to JLM. This study was also funded by the Fédération pour la Recherche sur le Cerveau in the form of a grant awarded to MLP and JLM. This study was also funded by the Academy of Finland (276612), Emil Aaltosen Säätiö Foundation, and Jalmari ja Rauha Ahokkaan Säätiö Foundation in the form of grants awarded to ASU and in the form of grants from IDEX Paris Saclay and INSERM - APHP 2010 interface grant awarded to HV.

**Competing interests:** The authors have declared that no competing interests exist.

**Abbreviations:** AAL, Anatomical Automatic Labeling; AUDIT, Alcohol Use Disorders Identification Test; CANTAB, Cambridge Automated Neuropsychological Test Battery; FWE, Family Wise Error; MNI, Montreal Neurologic Institute; mPFC, Medial Prefrontal Cortex; MPRAGE, Magnetisation Prepared Rapid Acquisition Gradient Echo; ROI, Region Of Interest; SDQ, Strengths and Difficulties Questionnaire; SPM, Statistical Parametric Mapping; VBM, Voxel-Based Morphometry; WD, Weekday; WE, Weekend.

habits, assessed with self-reports, and regional grey matter volumes, and psychological functioning, assessed with the Strengths and Difficulties Questionnaire and tests on working memory and impulsivity. Later weekend wake-up times were associated with smaller grey matter volumes in the medial prefrontal cortex and the amygdalae, and greater weekend delays in wake-up time were associated with smaller grey matter volumes in the right hippocampus and amygdala. The medial prefrontal cortex region mediated the correlation between weekend wake up time and externalising symptoms. Paying attention to regular sleep habits during adolescence could act as a protective factor against the emergence of psychopathology via enabling favourable brain development.

## Introduction

Sleep problems and psychiatric disorders increase sharply hand in hand during adolescence, but our understanding of the potential neurobiological links between them is only emerging [1]. Late sleep, sleep deprivation, and a delay of sleep timing (i.e. later bed and wake up times) on the weekend as compared to weekdays) have all been associated with a broad range of negative mental health consequences, including increased depressive and anxiety symptoms, increased risk-taking behaviours, as well as deteriorated executive functions [2–4]. Furthermore, sleep disturbances seem to precede the onset of diverse psychiatric disorders [5].

These studies support the theory that unhealthy sleep habits could affect the developing adolescent brain structure and thereby increase the vulnerability to various kinds of psychopathologies, but few studies on the relationship between adolescents' sleep habits and brain grey matter volumes have been published to date. In a sample of maltreated teenagers, reduced sleep efficiency was recently found to correlate with reduced grey matter volume (GMV) in hippocampus, inferior frontal gyrus and insula, suggesting that sleep might mediate the negative impact of adverse life events on brain morphology [6]. In a mixed sample of children and adolescents, weekday time in bed was found to correlate with regional grey matter volumes of the bilateral hippocampi and the dorsolateral prefrontal cortex [7]. In our previous study of 14-year-old adolescents, we found late sleep during the weekend and short sleep during the week to be associated with smaller regional grey matter volumes, particularly in the medial prefrontal cortex (mPFC). In addition, there was a correlation between mPFC GMV and school performance [8]. Since sleep characteristics and brain morphology undergo constant changes through adolescence [9, 10], it is important to study their interconnections repeatedly at different points of development.

The mPFC exerts an inhibitory top-down control of subcortical structures [11]. Poor sleep and eveningness-prone or irregular sleep rhythms can negatively affect adolescents' emotion regulation, reward-related processing, and impulse inhibition by influencing the mPFC [12–14] as well as the amygdala and the hippocampus [15–19]. These structures have also been implicated in the etiology and maintenance of psychiatric disorders [20, 21]. Studying the effects of sleep especially on the mPFC, the amygdala, and the hippocampus would thus crucially contribute to understanding the development of psychopathology during adolescence. Our general hypothesis is that adolescents' sleep patterns affect brain regional grey matter volumes, which in turn lead to lower psychological functioning or even mild psychopathology. Understanding the trajectories that lead toward psychiatric disorders as early as possible in development would allow us to develop effective intervention and prevention strategies.

## Aims of the study

In this follow-up study we aimed to assess whether our previous findings on the correlation between adolescents' sleep habits and regional brain grey matter volumes in the mPFC remained present at the age of 16, and to extend these findings by including the hippocampus and amygdala brain regions and examining the relationships with psychological functioning.

# Materials and methods

## Participants

Participants were recruited from schools near Paris, France, based on their age and absence of any major somatic condition. Written consent was obtained from all subjects in this study. The study was approved by the regional ethics committee (Comité de Protection des Personnes [CPP] Ile-de-France 7). The adolescents participated in a larger multi-centre study (http://www.imagen-europe.com/en/the-imagen-study.php) [22] at age 14 (baseline), and were followed up at age 16. Only the French adolescents were assessed for their sleep habits and were thus eligible for this study. Details of the sample at baseline have been previously reported [8]. This study focuses on the sample at age 16, at which time point written informed assent and consent to study participation were obtained from a total of 138 adolescents and their parents, respectively. We excluded participants who did not complete the sleep questionnaires, those whose Magnetic Resonance brain images did not pass the quality control of the raw or the segmented images, participants with brain lesions, and those with marked alcohol consumption (alcohol use disorders identification test (AUDIT) total score >7 [23]). In this study, we present data from the remaining 101 adolescents (mean age = 16.83 years, SD = 0.61; 56 girls; Table 1). At the time of the study, none of the participants were followed in the psychiatric care system and all participants were attending school regularly.

## Sleep assessments

Sleep habits were assessed by asking the adolescents their usual bed times and wake up times during weekdays (WD) and weekends (WE). Time in bed was approximated by calculating the number of hours between bed time and wake up time, separately for WD and WE. WE delay in wake up time and bed time ("social jet lag") was defined as the difference in wake up and bed times between weekdays and weekends, respectively. Sleep debt was defined as the difference in time in bed between weekdays and weekends.

## Psychological functioning

Symptom assessment was performed using the Strengths and Difficulties Questionnaire (SDQ) [24], a child and adolescent self-report questionnaire used to identify internalising and externalising problems. It consists of 25 items, five items for each subscale: conduct problems, hyperactivity, emotional problems, peer problems, and prosocial behaviour. The items are scored 0 to 2, reflecting the answers "not true", "somewhat true" or "certainly true". The scores are then summed, generating five scale scores ranging from 0 to 10 with higher scores reflecting more problems in the first four scales or more prosocial behaviour in the last scale. In low-risk samples, conduct problems and hyperactivity are best combined into an 'externalising' subscale, and emotional problems and peer problems into an 'internalising' subscale [25].

Other behavioural measures included the Kirby Delay-Discounting Questionnaire [26], a monetary choice questionnaire assessing cognitive impulsivity through delay discounting by having participants choose between smaller immediately available rewards and larger delayed rewards, and the spatial working memory task, a subtest of the computer-administrated

**Table 1. Clinical, behavioral characteristics, and brain volumes in community adolescents at age 16.**

| Variable | | Mean or % | SD |
|---|---|---|---|
| **Demographic variables** | Age (years) | 16.83 | 0.61 |
| | Sex | 55% (n = 56) female | |
| **Sleep variables (N = 101)** | Wake up time WD | 7:03 | 0:43 |
| | Wake up time WE | 10:01 | 1:15 |
| | Bed time WD | 22:42 | 0:43 |
| | Bed time WE | 00:05 | 1:10 |
| | WE delay wake up time | 2:59 | 1:17 |
| | WE delay bed time | 1:23 | 1:04 |
| | Time in bed WD | 8:18 | 0:59 |
| | Time in bed WE | 9:56 | 1:00 |
| | Sleep debt | 1:40 | 0:59 |
| **Performance scores (N = 82)** | Delay discounting large amounts | -2.25 | 0.68 |
| | Delay discounting medium amounts | -2.03 | 0.73 |
| | Delay discounting geomean | -2.06 | 0.66 |
| | Spatial working memory | 7.59 | 7.80 |
| **Behavioural problems (N = 74)** | SDQ internalising | 4.34 | 2.40 |
| | SDQ externalising | 7.36 | 2.68 |
| **Global brain measures (N = 101)** | Total grey matter volume | 742.02 | 68.99 |
| | Total white matter volume | 477.22 | 49.28 |
| | Total CSF | 395.55 | 40.89 |
| | Volume scaling factor | 1.32 | 148.21 |

WD = weekday; WE = weekend; CSF = cerebrospinal fluid; SDQ = Strenghts and Difficulties Questionnaire.

Cambridge Automated Neuropsychological Test Battery (CANTAB) measuring executive functioning [27]. The spatial working memory test has been widely used in typically developing and clinical populations of children and adolescents [28]. The error score was used as the outcome variable.

## MRI data acquisition and processing

MRI was performed on a 3T scanner (Siemens Trio). High-resolution anatomical MR images were obtained using a standardised 3D T1-weighted magnetisation prepared rapid acquisition gradient echo (MPRAGE) sequence based on the ADNI protocol (http://adni.loni.usc.edu/methods/mri-analysis/mri-acquisition/). The parameters were as follows: repetition time = 2,300 ms, echo time = 2.8 ms, flip angle = 8˚, 256x256x170 matrix, 1.1x1.1x1.1 mm voxel size.

The images were processed using Statistical Parametric Mapping 8 (SPM 8) using Voxel-Based Morphometry (VBM) [29]. The "unified segmentation" algorithm was used to normalise and segment the T1-weighted images into grey matter, white matter and cerebrospinal fluid. Home-made tissue probability maps were used instead of the standard template of SPM based on fully grown and developed brains of adults, who have larger brain volumes. The modulated images were smoothed with a 10-mm full-width at half-maximum isotropic Gaussian kernel. Head size was measured by the volume scaling factor, which is based on the affine transformation performed during spatial normalisation (https://surfer.nmr.mgh.harvard.edu/fswiki/eTIV).

## Statistical analyses

The outcome measures for the Kirby Delay-Discounting Questionnaire were calculated using an automated calculator [30]. A logistic regression that allowed for continuous estimates of k [31], and a logarithmic transformation to normalise the distribution of k-values were applied. The geometric mean of the k-values is bounded by the lowest implied indifference k-value at which subjects chose the larger delayed reward and the highest indifference k-value at which the subject choses the immediate reward.

Group comparisons for socio-demographic and clinical data, and global brain volumes were performed within the framework of the general linear model (GLM) using R software (http://cran.r-project.org). Voxel-wise comparisons were carried out within the GLM framework of SPM8. Sleep variables were the main factors while age, sex and volume scaling factor were entered as confounding variables. Pearson partial correlations between sleep and psychologicial function variables were calculated with age and sex as confounder variables.

*A priori* masks were used to examine the hippocampus, amygdala, and medial prefrontal region with a region of interest (ROI) approach. The hippocampus and amygdala masks were created with the Anatomical Automatic Labeling (AAL) atlas in SPM. The medial prefrontal region was marked with a 10 mm radius from coordinates [–2, 34, –14] taken from our previous results on an analogous sample at age 14, which indicated a strong relationship between sleep habits and GMV in this region [8]. Subsequently, it was investigated whether psychometrics measures (spatial working memory, delay discounting, internalising and externalising problems) were associated with grey matter volumes in these regions of interest.

At the voxel-level, statistical significance was set to p<0.05 Family Wise Error (FWE) corrected for multiple comparisons. Brain locations were reported as x, y, and z coordinates in Montreal Neurologic Institute (MNI) space.

In addition, it was examined whether the sleep assessment variables that were significant in the ROI VBM analyses, were correlated with psychological functioning. Subsequently, causal mediation analyses were performed to determine whether the grey matter clusters could mediate the relation between sleep and psychological functioning variables. These analyses were performed if there was a significant relationship between sleep and psychological functioning variables and GMV of the same ROI. The mediation analyses were performed with an algorithm using a set of general linear models to derive the mediation and direct effects from the total effect [32]. The psychological functioning measures were entered as dependent factor, and sleep variables as independent factor within a linear regression model. For the mPFC ROI analysis, raw volume was extracted from the smoothed, normalized, and modulated images and entered as mediator variable. The hippocampus and amygdala volumes were extracted from the same images using the AAL masks instead of the significant clusters in order to better approximate the complete volumes. Volume scaling factor and age were entered as confounding variables. Gender was added for analyses without a gender interaction between sleep and psychological functioning scores. If there was a gender effect, we performed the mediation analysis separately for boys and girls. This mediation model was performed using 5000 Monte Carlo draws for nonparametric bootstrap. In causal mediation analysis, a significant mediating effect is defined as a 95% confidence interval that does not include zero.

## Results

### Participant characteristics

On average, boys had later WD bed times (boys = 22.53 ± 0.47, girls = 22.33 ± 0.33; F(99) = 2.41, p = 0.02) and WE wake up times (mean wake up time: boys = 10.23 ± 1:17, girls = 9:44 ± 1:10;

F(99) = 2.63, p = 0.01) as compared to girls. As shown in Table 1, there were no gender differences in any of the other sleep variables, nor in delay discounting, spatial working memory or internalising and externalising problems.

Clinically relevant externalising symptoms (SDQ score ≥10), were present in 15 out of 81 adolescents (18.3%) and 10 out of 81 (12.2%) showed clinically relevant internalising symptoms (score ≥8).

### Sleep and regional grey matter volumes

WE wake up time correlated negatively with GMV in the bilateral amygdalae and the mPFC (p = 0.011, p = 0.041, p = 0.020, resp.; Table 2, Fig 1). WE delay in wake up time correlated negatively with GMV in the right hippocampal region and the right amygdala (p = 0.025, p = 0.008, resp.; Table 2). No statistically significant correlations between WE bed time, WE delay in bed time, or sleep debt and regional grey matter volumes were found.

**Table 2. Grey matter volume correlations with sleep measures in community adolescents at age 16 using regions-of-interest.**

| Sleep measure | Amygdala | | Hippocampus | | mPFC | |
|---|---|---|---|---|---|---|
| | T-value | Peak (x,y,z) | T-value | Peak (x,y,z) | T-value | Peak (x,y,z) |
| Weekend wake up time | 3.51* | 21,-6,-12 | | | 3.11* | 2,42,-11 |
| | 3.04* | -12,-3,-15 | | | | |
| Weekend delay in wake up time | 3.23* | 20,-3,-12 | | | | |
| | 4.23** | 20,-25,-23 | | | | |

* = p<0.05,

** = p<0.01,

*** = p<0.001 FWE corrected.

- = no significant results, Montreal neurological Institute coordinates are given for the voxel of maximal statistical significance; analyses are covaried for volume scale factor, age, and gender.

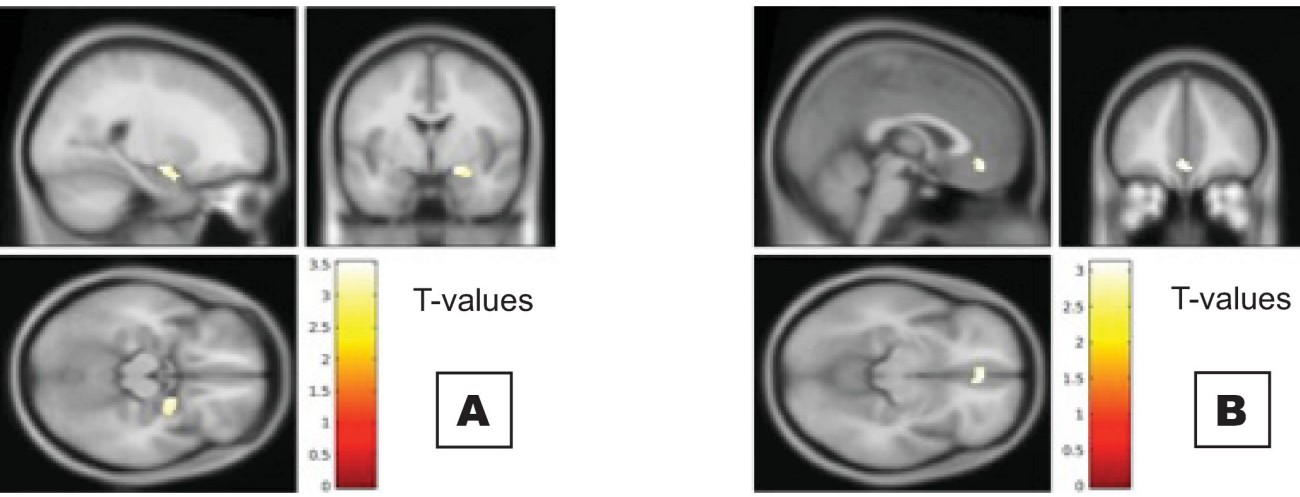

**Fig 1. Later wake up time during the weekend associated with reduced grey matter volumes in the (A) left amygdala and (B) medial prefrontal cortex (mPFC).** Corrected p FWE <0.05.

**Table 3. Correlations between sleep variables and psychological functioning in community adolescents at age 16 adjusted for age and sex.**

| | WE wake up time | WE delay wake up time |
|---|---|---|
| Delay discounting large | r = 0.02 | r = 0.07 |
| Delay discounting medium | r = 0.05 | r = 0.12 |
| Delay discounting geomean | r = 0.05 | r = 0.14 |
| SDQ externalising | r = 0.51*** | r = 0.61*** |
| SDQ internalising | r = -0.14 | r = -0.42** |
| Spatial working memory errors | r = -0.20 | r = 0.12 |

* = p<0.05,

** = p<0.01,

*** = p<0.001.

WE = weekend; SDQ = Strengths and Difficulties Questionnaire.

## Sleep and psychological functioning

Later WE wake up time and WE delay in wake up time correlated with measures of externalising problems (p<0.001), whilst a larger WE delay in wake up time was correlated with lower internalising problems (p<0.001) (Table 3).

## Grey matter volumes and psychological functioning

Smaller grey matter volumes in the mPFC region were associated with increased delay discounting (T = 3.28, p<0.01) as well as internalising (T = 4.99, p<0.001) and externalising (T = 6.03, p<0.001) problems (Table 4). Additionally, smaller grey matter volumes in the amygdala (most significant cluster T = 10.57, p<0.001), and hippocampal regions (T = 11.01, p<0.001) were associated with internalising problems. There were no other significant relations between grey matter volumes and psychological functioning measures (Table 4).

**Table 4. Grey matter volumes correlations with psychological functioning in 138 community adolescents at age 16.**

| Psychological functioning | Amygdala | | Hippocampus | | mPFC | |
|---|---|---|---|---|---|---|
| | T-value | x,y,z | T-value | x,y,z | T-value | x,y,z |
| Delay discounting large | - | - | - | - | 2.97* | -6,42,-15 |
| Delay discounting medium | - | - | - | - | 3.29* | -8,40,-17 |
| Delay discounting geomean | - | - | - | - | 3.28** | -8,40,-17 |
| Spatial Working Memory | - | - | - | - | - | - |
| SDQ externalising | - | - | - | - | 6.03*** | -2,44,-14 |
| SDQ internalising | 10.57*** | -30,-3,-29 | 11.01*** | -28,-7,-32 | 4.99*** | 4,30,-8 |
| | 4.40*** | 36,0,-24 | 7.34*** | 21,-45,-3 | | |
| | | | 6.13*** | 40,-15,-23 | | |

* = p<0.05,

** = p<0.01,

*** = p<0.001 FWE corrected.

- = no significant results, Montreal neurological Institute coordinates are given for the voxel of maximal statistical significance; analyses are covaried for volume scale factor,age, and sex. SDQ = Strengths and Difficulties Questionnaire.

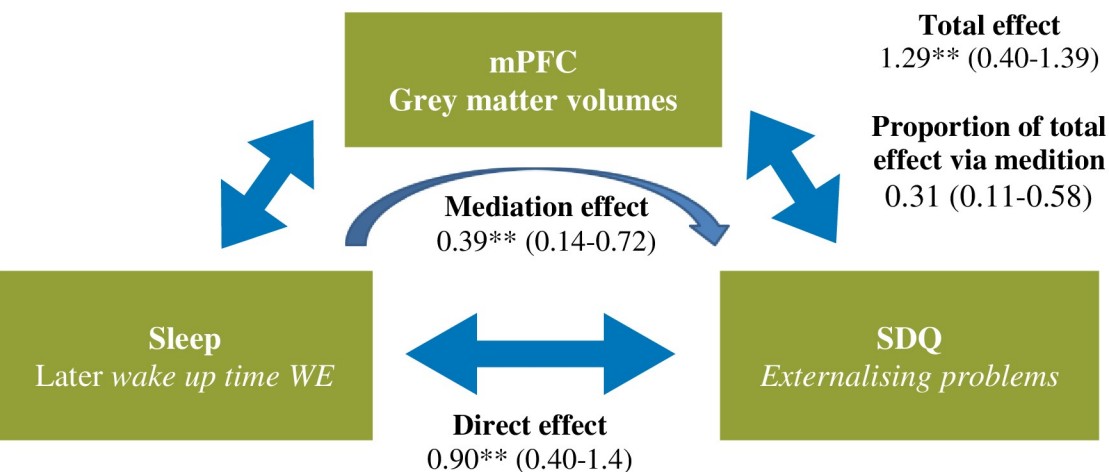

**Fig 2. Causal mediation analysis on the relationship between sleep variables and externalising problems with medial prefrontal cluster volumes as mediator.** Estimate of the size of the effect (95% confidence interval of the point estimate); * = $p<0.05$, ** = $p<0.01$, *** = $p<0.001$; SDQ = Strengths and Difficulties Questionnaire.

### Mediation analysis

Causal mediation analyses showed that variability in mPFC volumes accounted for 30.5% ($p<0.001$) of the total effect between wake up time during the WE and externalising problems (Fig 2).

## Discussion

The present findings confirm and extend our previous findings that sleep patterns in adolescents from the general population correlate with regional brain grey matter volumes and psychological functioning. In particular, later WE wake up times were associated with lower GMV in the mPFC and the amygdalae as well as with externalising symptoms. Smaller grey matter volumes in the mPFC region were associated with increased cognitive impulsivity as well as internalising and externalising problems. A causal mediation analyses showed that GMV of the mPFC mediated the relationship between weekend wake up time and externalising problems.

The results corroborate our previous findings in the sample at age 14, which showed an association between grey matter volumes in the mPFC region and weekend wake-up time [8]. This could indicate that weekend wake up times affect mPFC grey matter maturation throughout adolescence. The impact of sleep on the brain might be particularly significant and long-lasting during adolescence because the brain is still in development during this phase [33]. Since the mPFC is an important brain area for regulation of the limbic structures, it is plausible that impaired development of this structure due to unhealthy sleep habits can lead to cognitive and emotional control deficits [16, 18, 34]. Externalising symptoms and impulsivity are both related to a host of maladaptive behaviours, including drug abuse, gambling, and poor academic and work success [35, 36]. This is the first study to show a mediating role of grey matter volumes in the mPFC in the relationship between sleep and externalising symptoms in healthy adolescents.

The relationship between WE delay in wake up time and hippocampal and amygdala grey matter volumes could result from repeated challenges to the circadian regulatory mechanisms [37]. This can be compared with having regular *mini-jetlags*. A study on the short term effects of jetlag found reduced resting state activity in the mPFC and the left parahippocampal gyrus

as well as other default mode network regions [38]. A study on chronic functional jetlag reported a relationship between sleep patterns and right temporal lobe atrophy, which included the parahippocampal gyrus [39].

Smaller GMV in bilateral amygdala and parahippocampal regions were associated with more internalising problems. This finding is in accordance with previous research associating abnormalities in the amygdala, the parahippocampus, and the mPFC with emotional difficulties [40]. Internalising symptoms are predictors of later educational underachievement, mental disorders, and impaired personal relationships [41].

Interestingly, despite the association between a greater WE delay in wake up time and smaller hippocampal and amygdala volumes and the association of those reductions in GMVs with more internalising problems, a greater WE delay in wake up time was correlated with fewer internalising problems. There is evidence that sleep-wake rhythm disturbance is associated with better long-term outcome in depressed adolescents. This could be due to a factor that is independent of the GMVs examined here, for example a social factor, that causes a larger WE delay in wake up time and is beneficial for emotional wellbeing.

An alternative explanation to the relationship between later WE wake up times and larger WE delay in wake up times and GMVs is that later WE wake up times might reflect a late-prone biological rhythm. Chronotype has recently been found to correlate with local GMVs and cortical thickness in a small sample of adult men [42]. However, late bed times, another characteristic of a late chronotype, were not related to the volumes of the regions of interest in our study.

This is in contrast with our previous study in 14-year-old adolescents and a study by Taki and collaborators [7, 8], where weekend bed times were associated with smaller mPFC grey matter volumes, but the sample of these studies consisted of younger children, who have not yet developed a delay in circadian rhythms and whose time in bed is still much more regulated by the parents. Therefore, time in bed during the week might be a more important factor in earlier brain developmental phases, while sleep timing and regularity in sleep timing might be more important later during adolescence. The current results also do not provide evidence of negative effects of sleep debt, as measured by time in bed difference between WD and WE, on the adolescent brain. It must be noted, however, that sleep debt could affect brain regions that were not examined in the current study.

The findings could be confounded by the developmental stage of the participants. However, we used a relatively large sample, drawn from the general population, with a very narrow age range, making the results less likely to be influenced by confounding factors associated with age and sampling bias. A limitation of this study is the use of self-report questionnaires to measure sleep and behavioural problems. Sleep diaries and polysomnographic or actigraphic recordings would have provided a more objective measure of sleep, and including parental reports could have completed the picture of behavioural problems in adolescents. As this is a cross-sectional study, it is difficult to see to which degree sleep habits are a cause or a consequence of reduced grey matter volumes. We theorise that sleep rhythms influence brain development, which in turn causes lower functioning. The results of the causal mediation analyses would also favour this interpretation. However, reduced grey matter volumes might be a pre-existing condition that contributes to cognitive impulsivity and behavioural problems as well as the development of specific sleep patterns. Lastly, the ROI approach does not allow exploration of effects of sleep brain on regions other than the chosen ROIs.

## Conclusions

Overall, the present findings are consistent with and extend our previous report in 14 year-old adolescents, suggesting that the negative impact of a later weekend wake up time on the

adolescent brain can result in reduced ability to regulate emotions. This highlights the importance of sleep habits in adolescents and supports the recommendation to keep variability in sleep timing to a minimum in order to reduce the risk of psychiatric morbidity.

## Acknowledgments

Imagen Consortium collaborators: M Fauth-Bühler, L Poutska, F Nees, Y Grimmer, M Struve from Central Institute of Mental Health, Mannheim, Germany;

A Ströhle, V Kappel, B M van Noort, from Charité hospital, Berlin, Germany;

N Bordas, Z Bricaud, I Filippi, A Galinowski, F Gollier-Briant, Vincent Ménard, from INSERM, France;

A Cattrell, R Goodman, A Stringaris, C Nymberg, L Reed, from the Institute of Psychiatry, Psychology & Neuroscience, King's College London, United Kingdom;

B Ittermann, R Brühl R, from Physikalisch-TechnischeBundesanstalt (PTB), Berlin; Germany;

T Hübner, K Müller, from University of Dresden, Germany;

U Bromberg, J Gallinat, T Fadai, from University of Hamburg, Germany;

P Gowland, C Lawrence, from University of Nottingham, United Kingdom;

## Author Contributions

**Conceptualization:** Anna S. Urrila.

**Writing – original draft:** Anna S. Urrila.

**Writing – review & editing:** Winok Lapidaire, Anna S. Urrila, Eric Artiges, Ruben Miranda, Hélène Vulser, Pauline Bézivin-Frere, Hervé Lemaître, Jani Penttilä, Tobias Banaschewski, Arun L. W. Bokde, Uli Bromberg, Christian Büchel, Patricia J. Conrod, Sylvane Desrivières, Vincent Frouin, Jürgen Gallinat, Hugh Garavan, Penny Gowland, Andreas Heinz, Bernd Ittermann, Dimitri Papadopoulos-Orfanos, Tomáš Paus, Michael N. Smolka, Gunter Schumann, Marie-Laure Paillère Martinot, Jean-Luc Martinot.

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
