## [Decision Letter · Decision Letter 0]

8 Jun 2020

PONE-D-19-34596

Sleep, regional grey matter volumes, and psychological functioning in adolescents

PLOS ONE

Dear Dr. Lapidaire,

Thank you for submitting your manuscript to PLOS ONE. After careful consideration, we feel that it has merit but does not fully meet PLOS ONE’s publication criteria as it currently stands. Therefore, we invite you to submit a revised version of the manuscript that addresses the points raised during the review process.

Although this manuscript is generally well-written and technically sound, it has several methodological issues, particularly for the statistical analysis, that should be clearly resolved.

We look forward to receiving your revised manuscript.

Kind regards,

Kyoung-Sae Na, M.D.

Academic Editor

PLOS ONE

Journal Requirements:

3. One of the noted authors is a group or consortium [IMAGEN consortium]. In addition to naming the author group, please list the individual authors and affiliations within this group in the acknowledgments section of your manuscript. Please also indicate clearly a lead author for this group along with a contact email address

Additional Editor Comments (if provided):

Reviewers' comments:

Reviewer's Responses to Questions

**Comments to the Author**

1. Is the manuscript technically sound, and do the data support the conclusions?

Reviewer #1: Yes

Reviewer #2: Yes

2. Has the statistical analysis been performed appropriately and rigorously? 

Reviewer #1: No

Reviewer #2: Yes

3. Have the authors made all data underlying the findings in their manuscript fully available?

Reviewer #1: No

Reviewer #2: Yes

4. Is the manuscript presented in an intelligible fashion and written in standard English?

Reviewer #1: Yes

Reviewer #2: Yes

5. Review Comments to the Author

Reviewer #1: Although this study was not well controlled as all sleep characteristics were obtained by self-report, the research question is interesting and important. Generally, the authors found that the delayed weekend wake-up time is associated with the reduced regional GMV, and the psychological functioning as well. The findings can provide some insights in adolescents’ sleep habit. However, I have some concerns of the analyses and results that will need to be addressed or clarified before it can be published.

Major concerns:

1. For the whole brain analyses (i.e. Figure 1, Table 2), it was reported that the threshold is Pfwe < 0.05, but was it the whole-brain level multiple comparison correction or just cluster-level? If it is cluster-level, what is the whole brain threshold. Please clarify.

2. No Figure caption of Figure 1. And why do not you present the whole brain correlation results between variability wake-time and Amygdala, Hippocampus?

3. I am a little confused that you applied different approaches to explore the GMV and wakeup-time and the psychological functioning. I understand you have hypotheses in those regions (i.e. hippocampus, amygdala, mPFC). But since you found the correlation between the wakeup-time and regional GMV in those regions at the whole-brain level, why didn’t you apply the same approach to explore the relationship between the GMV and the psychological functioning, especially you found the correlation between wake-time and the functioning.

4. According to your method, you did the ROI approach to explore the GMV and psychological functioning, and for the mPFC, you defined the ROI based on your previous paper, the coordinate is [-2, 34 ,-14]. However, when you reported the correlation results in Table4, apparently ,the coordinate is not the one you pre-defined. Have you applied whole-brain analyses here? I am really confused about it. Refer to the comment above, this analysis and results need be addressed or clarified.

5. Sleep debt is an important factor. Although you mentioned it as a limitation that you did not investigate it. I am curious what prevent you from exploring it? Since it will make the results stronger.

6. For the causal mediation analyses, it is interesting. But it could be more clear if you can present the mediation figures with all variables. Also, I cannot see any causal direction from the result, and you did not discuss this result at all.

Minor issues:

7. The concepts throughout the manuscripts are inconsistent and not clear. Is the WE delay in wake-up time in the abstract the same thing as variability of wake-up time in Table 2?

8. In the Sleep assessments, it says “ WE delay in sleep timing (“social jet lag”) and weekly sleep debt was defined as the difference between weekday and weekend in sleep times and time in bed.” What is sleeping timing & sleep times here? Is the sleep onset or time go to bed?

9. The first sentence in the 4th paragraph of statistical analyses section is very confused. “In addition, it was examined whether the sleep times that were significant in the ROI VBM analyses were correlated with psychological functioning.” I do not understand what analyses you’ve done here. And I do not think you have done any analyses about “sleep times” in the current manuscript?

10. The title is too broad that you say sleep, it is really just about WE wake-up time in the current study.

Reviewer #2: Title: Sleep, regional grey matter volumes, and psychological functioning in adolescent

Manuscript Number: PONE-D-19-34596

This manuscript deals with an interesting topic and is well written. I expect that if this manuscript is being published as an article in the PLOS ONE, it will draw a lot of attention from psychiatrists.

Although this study has several limitations (e.g. the cross-sectional nature, the use of a self-report sleep scale instead of an objective measure for sleep), the authors addressed these limitations well. Also, the authors tried to minimize potential confounding factors such as developmental effect or gender effect through the use of statistical adjustment or subgroup analysis.

Minor concerns

1. Was there any interaction effect between gender and sleep-related variables or gender and psychological functioning scores? Please present it in the result section if there was any (Line 252-255).

2. Please provide explanations for acronyms when first presented in the text (e.g., line 115)

3. Please provide p values throughout the text.

4. Please add an explanation for the acronym SDQ as a footnote in the tables (table 1, 3 and 4).

6. PLOS authors have the option to publish the peer review history of their article (what does this mean?). If published, this will include your full peer review and any attached files.

Reviewer #1: No

Reviewer #2: Yes: Min-Hyeon Park

---

## [Author Response · Author response to Decision Letter 0]

23 Oct 2020

Response to Reviewers

We thank the reviewers for their thorough and inquisitive review. We believe the paper has greatly benefited from their input. Please find below the responses to the points raised.

Reviewer #1:

Although this study was not well controlled as all sleep characteristics were obtained by self-report, the research question is interesting and important. Generally, the authors found that the delayed weekend wake-up time is associated with the reduced regional GMV, and the psychological functioning as well. The findings can provide some insights in adolescents’ sleep habit. However, I have some concerns of the analyses and results that will need to be addressed or clarified before it can be published.

Major concerns:

1. For the whole brain analyses (i.e. Figure 1, Table 2), it was reported that the threshold is Pfwe < 0.05, but was it the whole-brain level multiple comparison correction or just cluster-level? If it is cluster-level, what is the whole brain threshold. Please clarify.

• No whole-brain analysis was performed for this paper, since we tested priors on specific brain regions (ROIs). The FWE p-value was corrected for multiple comparisons within the ROIs’ voxels. 

This conservative method is acknowledged as there is no universal definition for statistical thresholds (analogous issues have been raised for fMRI : Eklund A, Nichols TE, Knutsson H. 2016. Cluster failure: Why fMRI inferences for spatial extent have inflated false-positive rates. Proc Natl Acad Sci USA 113:7900–7905. Erratum in Proc Natl Acad Sci USA 113:E4929.)

2. No Figure caption of Figure 1. And why do not you present the whole brain correlation results between variability wake-time and Amygdala, Hippocampus?

• Corrected, Figure 1 now has a caption.

• No whole-brain analysis was performed for this paper, since we tested a priori hypotheses on specific brain regions that according to the literature and our previous study could be particularly affected by sleep variables and psychological functioning measures. 

3. I am a little confused that you applied different approaches to explore the GMV and wakeup-time and the psychological functioning. I understand you have hypotheses in those regions (i.e. hippocampus, amygdala, mPFC). But since you found the correlation between the wakeup-time and regional GMV in those regions at the whole-brain level, why didn’t you apply the same approach to explore the relationship between the GMV and the psychological functioning, especially you found the correlation between wake-time and the functioning.

• As mentioned above, no whole-brain analysis was performed. The approach to explore relationships between GMV - sleep and GMV – psychological functioning is exactly the same. The results were presented slightly differently, because there were many results to present both cluster and peak statistics for the psychological functioning. We completely agree that this is confusing and have therefore adapted table 2 to be in the same format as table 4.

4. According to your method, you did the ROI approach to explore the GMV and psychological functioning, and for the mPFC, you defined the ROI based on your previous paper, the coordinate is [-2, 34 ,-14]. However, when you reported the correlation results in Table4, apparently ,the coordinate is not the one you pre-defined. Have you applied whole-brain analyses here? I am really confused about it. Refer to the comment above, this analysis and results need be addressed or clarified.

• The coordinates in Table 4 show the coordinates of the voxel of maximal statistical significance. This point is within the ROI, but not exactly in the centre of the ROI (the [-2, 34 ,-14] for the mPFC). We did not perform whole-brain analysis here. We put the line “Montreal neurological Institute coordinates are given for the voxel of maximal statistical significance” in the footnotes of all relevant tables to avoid confusion.

5. Sleep debt is an important factor. Although you mentioned it as a limitation that you did not investigate it. I am curious what prevent you from exploring it? Since it will make the results stronger.

• Sleep debt (as defined as the difference in time in bed between weekdays and weekends) was investigated, but did not show a relationship with the regional grey matter volumes that were the focus in this study (as mentioned in the 6th paragraph of the discussion). This could have been mentioned more clearly in the results section, so we made this more explicit in the “Sleep and regional grey matter volumes” paragraph of the results section. 

6. For the causal mediation analyses, it is interesting. But it could be more clear if you can present the mediation figures with all variables. Also, I cannot see any causal direction from the result, and you did not discuss this result at all.

• Excellent point, adding a figure has made it much clearer. At closer examination, the mediation analysis of WE wake up time, mPFC, and internalising problems did not meet the pre-set requirements (because WE wake up time was not significantly associated with internalising problems) and was therefore removed. Causal mediation analysis (CMA) is a method to dissect total effect into direct and indirect effect. The indirect effect is transmitted via mediator to the outcome. Although this terminilogy is commonly used, we agree with the referee that no actual cause is demonstrated by this method, besides priors on the variables selected for CMA and directions of the mediation. 

For your reference, the statistical methods section states “Subsequently, causal mediation analyses were performed to determine whether the grey matter clusters could mediate the relation between sleep and psychological functioning variables. These analyses were performed if there was a significant relationship between sleep and psychological functioning variables and GMV of the same ROI.”

Minor issues:

7. The concepts throughout the manuscripts are inconsistent and not clear. Is the WE delay in wake-up time in the abstract the same thing as variability of wake-up time in Table 2?

• Yes. We agree this is confusing and therefore we clarified the concepts in the section “Sleep assessments” and changed the naming in Table 2.

8. In the Sleep assessments, it says “ WE delay in sleep timing (“social jet lag”) and weekly sleep debt was defined as the difference between weekday and weekend in sleep times and time in bed.” What is sleeping timing & sleep times here? Is the sleep onset or time go to bed?

• We clarified the concepts in the section “Sleep assessments”. Sleep timing and sleep times (the time at which the participants went go to bed and woke up) were used interchangeably, but since this can indeed be confusing, we have now only used the term sleep timing. 

9. The first sentence in the 4th paragraph of statistical analyses section is very confused. “In addition, it was examined whether the sleep times that were significant in the ROI VBM analyses were correlated with psychological functioning.” I do not understand what analyses you’ve done here. And I do not think you have done any analyses about “sleep times” in the current manuscript?

• We changed the wording in the manuscript to clarify. This analysis is to assess whether sleep relates to psychological functioning, but we have limited ourselves to only investigate those sleep variables that showed significant relationships to the grey matter volumes. These results of these analyses are presented in table 3.

10. The title is too broad that you say sleep, it is really just about WE wake-up time in the current study.

• We changed the title to: Irregular sleep habits, regional grey matter volumes, and psychological functioning in adolescents

Reviewer #2: 

This manuscript deals with an interesting topic and is well written. I expect that if this manuscript is being published as an article in the PLOS ONE, it will draw a lot of attention from psychiatrists.

Although this study has several limitations (e.g. the cross-sectional nature, the use of a self-report sleep scale instead of an objective measure for sleep), the authors addressed these limitations well. Also, the authors tried to minimize potential confounding factors such as developmental effect or gender effect through the use of statistical adjustment or subgroup analysis.

Minor concerns

1. Was there any interaction effect between gender and sleep-related variables or gender and psychological functioning scores? Please present it in the result section if there was any (Line 252-255).

2. Please provide explanations for acronyms when first presented in the text (e.g., line 115)

• We added the full name in line 115 and double checked that all acronyms in text were spelled out when first presented.

3. Please provide p values throughout the text.

• We have added p-values in-text to all results sections

4. Please add an explanation for the acronym SDQ as a footnote in the tables (table 1, 3 and 4).

• Thank you for this suggestion, we have added this to the table footnotes.

In addition to addressing the reviewers’ comments, we have made some additional changes to improve the manuscript.

• Since we adjusted the grey matter volume analyses for sex and age, we thought it best to also adjust for these factors in the direct correlation between sleep variables and psychological functioning. We thus performed Pearson partial correlation analyses. We have updated the tables, the text descriptions, and the methods section.

• We added reflections on the negative correlation between WE delay wake up time and internalising problems in the discussion.

• “At closer examination, the mediation analysis of WE wake up time, mPFC, and internalising problems did not meet the pre-set requirements (because WE wake up time was not significantly associated with internalising problems) and was therefore removed”.

---

## [Decision Letter · Decision Letter 1]

30 Nov 2020

Irregular sleep habits, regional grey matter volumes, and psychological functioning in adolescents

PONE-D-19-34596R1

Dear Dr. Martinot,

We’re pleased to inform you that your manuscript has been judged scientifically suitable for publication and will be formally accepted for publication once it meets all outstanding technical requirements.

Kind regards,

Kyoung-Sae Na, M.D.

Academic Editor

PLOS ONE

Additional Editor Comments (optional):

Reviewers' comments:

Reviewer's Responses to Questions

**Comments to the Author**

1. If the authors have adequately addressed your comments raised in a previous round of review and you feel that this manuscript is now acceptable for publication, you may indicate that here to bypass the “Comments to the Author” section, enter your conflict of interest statement in the “Confidential to Editor” section, and submit your "Accept" recommendation.

Reviewer #1: All comments have been addressed

Reviewer #2: All comments have been addressed

2. Is the manuscript technically sound, and do the data support the conclusions?

Reviewer #1: Yes

Reviewer #2: Yes

3. Has the statistical analysis been performed appropriately and rigorously? 

Reviewer #1: Yes

Reviewer #2: Yes

4. Have the authors made all data underlying the findings in their manuscript fully available?

Reviewer #1: No

Reviewer #2: Yes

5. Is the manuscript presented in an intelligible fashion and written in standard English?

Reviewer #1: Yes

Reviewer #2: Yes

6. Review Comments to the Author

Reviewer #1: Thank you for considering and addressing all comments. The authors have answered all my questions well. The current manuscript can be accepted.

Reviewer #2: The authors have systematically addressed or clarified my major concerns. Moreover, the manuscript is clear. The authors have answered the suggestions of the reviewers point by point. The revised version is much better. I recommend acceptance.

7. PLOS authors have the option to publish the peer review history of their article (what does this mean?). If published, this will include your full peer review and any attached files.

Reviewer #1: No

Reviewer #2: **Yes: **Min-Hyeon Park

---

## [Editor Report · Acceptance letter]

22 Dec 2020

PONE-D-19-34596R1 

Irregular sleep habits, regional grey matter volumes, and psychological functioning in adolescents 

Dear Dr. Martinot:

I'm pleased to inform you that your manuscript has been deemed suitable for publication in PLOS ONE. Congratulations! Your manuscript is now with our production department. 

Kind regards, 

on behalf of

Dr. Kyoung-Sae Na 

Academic Editor

PLOS ONE